# Identifying A(s) and β(s) in Single-Loop Feedback Circuits Using the Intermediate Transfer Function Approach

**DOI:** 10.3390/s22114303

**Published:** 2022-06-06

**Authors:** Gordon Walter Roberts

**Affiliations:** Integrated Microsystems Laboratory, Department of Electrical and Computer Engineering, McGill University, Montreal, QC H3A 0G4, Canada; gordon.roberts@mcgill.ca

**Keywords:** negative feedback circuits, single-loop feedback topologies, loop transmission function, closed-loop operation, return ratio, intermediate transfer functions

## Abstract

It is common practice to model the input–output behavior of a single-loop feedback circuit using the two parameters, **A** and **β**. Such an approach was first proposed by Black to explain the advantages and disadvantages of negative feedback. Extensive theories of system behavior (e.g., stability, impedance control) have since been developed by mathematicians and/or control engineers centered around these two parameters. Circuit engineers rely on these insights to optimize the dynamic behavior of their circuits. Unfortunately, no method exists for uniquely identifying **A** or **β** in terms of the components of the circuit. Rather, indirect methods, such as the injection method of Middlebrook or the break-the-loop approach proposed by Rosenstark, compute the return ratio ***RR*** of the feedback loop and inferred the parameters **A** and **β**. While one often assumes that the zeros of (1 + ***RR***) are equal to the zeros of (1 + **A** × **β**), i.e., the closed-loop poles are equivalent, this is not true in general. It is the objective of this paper to present an exact method to uniquely identify each feedback parameter, **A** or **β**, in terms of the circuit components. Further, this paper will identify the circuit conditions for which the product of **A** × **β** leads to the correct closed-loop poles.

## 1. Introduction

Circuit designers make use of single-loop negative feedback techniques to improve the robustness of their circuits. By feeding back a portion of the output signal to subtract from the input signal before applying it to the main circuit creates a circuit that is more robust to changes in the main circuit behavior, noise on the power supply and other practical concerns. As first described by Black [1], with the main circuit designated as an **A**(s) block, and the feedback circuit designated with a **β**(s) block, the arrangement of the two circuits can then be described as that shown in Figure 1. The critical observation made by Black at the time was the idea of a subtraction operation that combined the input with the feedback signal.

While using a block diagram to describe the operation of a circuit is a very popular method for describing the input–output behavior of circuits, there does not exist any method that can be used to uniquely identify **A**(s) and **β**(s) in terms of the circuit components, e.g., resistors, capacitors, etc. There does exist, however, many methods [2,3,4,5,6,7,8,9,10] that can be used to extract the return ratio ***RR***(s), i.e., inverted gain around a feedback loop [11].

By assuming this is equivalent to the product of **A**(s) × **β**(s), known as the loop transmission or loop gain, one can factor this into two separate terms, **A**(s) and **β**(s). Two problems exist with this approach: (1) the return ratio ***RR***(s) may not be equal to the **A**(s) × **β**(s), and (2) factoring **A**(s) × **β**(s) into two separate terms, **A**(s) and **β**(s), may not be so obvious.

In this paper, a general method for identifying the feedback parameters **A**(s) and **β**(s) for any single-loop negative feedback circuit will be described. The method is based on the concept of intermediate transfer functions (IFs) [12,13]. The proposed method can be performed using any Spice-like program, as no additional tools are required, or simply worked by hand or on a computer using traditional circuit analysis techniques. A distinction is made between single-loop feedback circuits that conform to the structure proposed by Black and those that do not. This is equivalent to identifying the conditions when the return ratio ***RR***(s) will predict the same closed-loop poles as those predicted by the product of **A**(s) × **β**(s). In more mathematical terms, using the language of Nyquist [14], this can be stated as:(1)$$\mathrm{zeros}\left\{1+\mathrm{RR}\left(s\right)\right\}=\mathrm{zeros}\left\{1+A\left(s\right)\times \beta \left(s\right)\right\}$$

While it is common practice to infer from Equation (1) that the return ratio ***RR***(s) is equivalent to the product of **A**(s) × **β**(s), i.e., $\mathrm{RR}\left(s\right)=A\left(s\right)\times \beta \left(s\right)$, this is not generally true. Instead, Equation (1) suggests the following more general equivalency as developed in Appendix A, as
(2)$$\mathrm{num}\left\{\mathrm{RR}\left(s\right)\right\}+\mathrm{den}\left\{\mathrm{RR}\left(s\right)\right\}=\mathrm{num}\left\{A\left(s\right)\times \beta \left(s\right)\right\}+\mathrm{den}\left\{A\left(s\right)\times \beta \left(s\right)\right\}$$
where ***num***{+} and ***den***{+} are the numerator and denominator polynomial terms of its argument between the curly brackets. Consequently, any method used to extract the return ratio ***RR***(s) cannot be used to uniquely identify the single-loop feedback parameters **A**(s) or **β**(s). Thus, highlighting the need for this work.

The paper will begin in Section 2 by describing the block diagram of a single-loop negative feedback structure first proposed by Black [1]. Section 3 describes the four network topologies of a single-loop feedback circuit and the circuit conditions required to be *compliant* with the single-loop feedback structure proposed by Black. In other words, when the return ratio and product **A**(s) × **β**(s) satisfy Equation (1). Section 4 describes a general circuit analysis method for identifying the feedback parameters **A**(s) and **β**(s). It is based on defining a set of intermediate transfer functions (IFs) from the input to the feedback variables of a circuit and relating them to the feedback parameters. Section 5 will provide a method that applies to any circuit that implements a single-loop feedback arrangement, but one that is *noncompliant* with the single-loop feedback structure described by Black. In other words, when the return ratio ***RR***(s) and **A**(s) × **β**(s) does not satisfy Equation (1). An example will be provided to highlight these two distinctions in Section 6. There are several ways in which to identify the error and feedback signals of the front-end mixing process, as well as the variable that the loop is sensing. In other words, the feedback parameters **A**(s) and **β**(s) are not unique. This is highlighted in Section 7, using a two-stage BJT amplifier with a resistive feedback circuit example. Subsequently, in Section 8, the invariance property of any feedback formulation drawn from the same circuit is described. Finally, conclusions are drawn in Section 9.

## 2. The Single-Loop Negative Feedback Structure

The basic structure of a system, including some forms of negative feedback as proposed by Black, is shown in the block diagram form in Figure 1. The input signal is defined by the variable $x_{s}$(s) and the output signal as $x_{o}$(s). Here, the block depicted by **A**(s) represents the feedforward gain stage of the closed-loop system, and the block denoted by **β**(s) represents the feedback block. It is assumed that the signal propagates in the direction of the arrows through these building blocks, i.e., they are unilateral. There is no signal that propagates back through the block. The summing node is used to subtract the *feedback signal* ($x_{Fdbk}$(s)) that is fed back from the output signal from the input signal $x_{s}$(s) to create what is known as the *error signal* $x_{\mathrm{Err}}$(s) being
(3)$$x_{\mathrm{Err}}\left(s\right)=x_{s}\left(s\right)-x_{Fdbk}\left(s\right)$$

Equation (3) is the central equation that all single-loop negative feedback circuits must implement. A circuit which does not implement this equation cannot be classified as a single-loop negative feedback circuit, as first proposed by Black.

Both **A**(s) and **β**(s) can be expressed in terms of these intermediate signals, $x_{Err}$(s) and $x_{Fdbk}$(s), together with the output variable $x_{o}$(s). For instance, the feedforward block **A**(s) can be defined as:(4)$$A\left(s\right)=\frac{x_{o}\left(s\right)}{x_{\mathrm{Err}}\left(s\right)}$$
and the feedback block **β**(s) can be written as:(5)$$\beta \left(s\right)=\frac{x_{\mathrm{Fdbk}}\left(s\right)}{x_{o}\left(s\right)}.$$

Consequently, Equation (3) can be re-written in terms of the feedback block **β**(s) as:(6)$$x_{\mathrm{Err}}\left(s\right)=x_{s}\left(s\right)-\beta \left(s\right)x_{o}\left(s\right).$$

Substituting Equation (6) into (4) one can write the input–output transfer function **A*_f_***(s) of the overall closed-loop system as:(7)$$A_{f}\left(s\right)=\frac{x_{o}\left(s\right)}{x_{s}\left(s\right)}=\frac{A\left(s\right)}{1+A\left(s\right)\beta \left(s\right)}.$$

## 3. The Four Basic Feedback Amplifier Topologies

Amplifiers incorporating a single-loop negative feedback loop can be divided into four general classes depending on the nature of the sensing signal (i.e., voltage or current) and how the feedback signal combines or *mixes* with the input signal, and on how the feedback signal is *sensed*. The four general classes are described here as follows: (i) voltage-mixing/voltage-sensing, (ii) voltage-mixing/current-sensing, (iii) current-mixing/voltage-sensing, and (iv) current-mixing/current-sensing. Others have used descriptions, such as series-series or shunt-series for the mixing and sensing operation [4]. As these terms are less explicit, they will not be used here.

Figure 2 illustrates the four circuit topologies that are used to realize a single-loop negative-feedback circuit. Figure 2a illustrates the voltage-mixing/voltage-sensing topology where the input and output signals are $v_{s}$ and $v_{o}$. As the input voltage signal is connected in series with the input to the amplifier and the output signal from the feedback network, the front-end portion of this topology is said to implement a voltage-mixing, or series connection. At the output, the amplifier generates a voltage signal $v_{o}$ and this signal is “sensed” by the feedback amplifier to generate the feedback signal. Thus, the overall topology is referred to as a voltage-mixing/voltage-sensing arrangement. A second topology is shown in Figure 2b. This topology has the same front-end arrangement where the input signal is connected in series with the input to the amplifier and the output of the feedback circuit. As the output is a current signal $i_{o}$, and it is this signal that is “sensed” by the feedback network, this topology is referred to as a voltage-mixing/current-sensing arrangement. In contrast, the two topologies of Figure 2c,d use a current source as excitation and have the amplifier input placed in parallel with the output of the feedback network. One refers to this arrangement as a current-mixing one. In the case of the topology shown in Figure 2c, the output signal is a current signal. Thus, the feedback network “senses” the output current. Therefore, the topology of Figure 2c is said to be a current-mixing/current-sensing arrangement. Finally, the topology of Figure 2d generates a voltage as its output signal and the feedback network senses this quantity. Consequently, the topology of Figure 2d is called a current-mixing/voltage-sensing arrangement.

### 3.1. Input Signal Mixing Compliance

The four topologies of Figure 2 have an important limitation when it comes to realizing the single-loop negative feedback structure of Figure 1. To understand this, consider the voltage-mixing arrangement shown in Figure 3a. If one assumes the signal fed back by the feedback network is the signal $v_{Fdbk}$ and the signal driving the amplifier is $v_{Err}$, then according to KVL around the loop formed at the input, one can write:(8)$$v_{\mathrm{Err}}=v_{s}-v_{\mathrm{Rs}}-v_{\mathrm{Fdbk}}.$$

If one assumes a one-to-one correspondence between the signals of the topology of Figure 3a and that of the feedback structure proposed by Black, as shown in Figure 1, one can write:(9)$$\begin{array}{c} v_{s}\\ v_{\mathrm{Err}}\\ v_{\mathrm{Fdbk}\;\;\;\;} \end{array}\begin{array}{c} ↔\\ ↔\\ ↔ \end{array}\begin{array}{c} x_{s}\\ x_{\mathrm{Err}}\\ \;\;\;x_{\mathrm{Fdbk}} \end{array}$$

The voltage variable $v_{Rs}$ is clearly not accounted for by Black’s theory. Thus, the error signals defined by Equations (3) and (8) are different. The topology of Figure 3a is not compliant with the mixing arrangement proposed by Black.

Likewise, a similar result occurs with the current-mixing topology highlighted in Figure 3b. Here, one can use KCL and relate the current signals at the input port of the amplifier as:(10)$$i_{\mathrm{Err}}=i_{s}-i_{Rs}-i_{\mathrm{Fdbk}}$$

Assuming the following one-to-one correspondence between Figure 1 and Figure 3b, one can claim:(11)$$\begin{array}{c} i_{s}\\ i_{\mathrm{Err}}\\ i_{\mathrm{Fdbk}\;\;\;\;} \end{array}\begin{array}{c} ↔\\ ↔\\ ↔ \end{array}\begin{array}{c} x_{s}\\ x_{\mathrm{Err}}\\ \;\;\;x_{\mathrm{Fdbk}} \end{array}$$
and conclude that the current variable $i_{Rs}$ is not accounted for by Black’s theory. Thus, the error signals defined by Equations (3) and (10) are different. The topology of Figure 3b is not compliant with the structure proposed by Black.

Fortunately, there is a simple topological fix that can be used to ensure a single-loop negative feedback circuit is compliant with the mixing arrangement proposed by Black. By associating the source resistance $R_{s}$ with either the basic amplifier or the feedback network, the feedback circuit can be made compliant with the feedback structure proposed by Black while maintaining circuit equivalence. These situations are depicted in Figure 4 and Figure 5 for the voltage and current-mixing arrangements. As is evident from these two figures, feedback compliance comes down to selecting the error and feedback signals appropriately.

### 3.2. Output Signal Sensing Compliance

Voltage or current sensing refers to the signal that is being sensed and fed back to the mixing element through the feedback network. The circuit variable which the circuit senses is either a node voltage or branch current, as the choice is somewhat arbitrary. However, it is paramount that the sense variable is located directly on the circuit path of the feedback loop as either a node voltage or branch current. Any circuit variable not on the path of the feedback loop cannot be sensed and be used to provide corrective action. In many circumstances, the sense variable is not equal to the designated output. For instance, in Figure 6a the output of the circuit with a feedback loop is designated as the output voltage $v_{o}$. As the feedback network is connected in parallel with the amplifier output, the sense signal is the voltage developed across the load resistor $R_{L}$; this is also designated as the output voltage $v_{o}$. The arrangement shown in Figure 6a would therefore be sense compliant with what Black intended. In the situation depicted in Figure 6b, the output of the circuit is again designated as the voltage across the load $R_{L}$, however, the feedback network is connected in series with the load resistor and will be sensing the load current $i_{L}$ instead. Consequently, according to Black’s theory, the sense variable should be designated as the load current. In other words, the arrangement of Figure 6b would not be sense compliant with Black’s intention. To avoid any future confusion, we will designate the output variable of the feedback topology proposed by Black as the sense variable and designate it as $x_{Sen}$(s). As this signal may be different from the output designated signal variable, an additional block denoted by ***γ***(s) is included, as shown in Figure 7. Consequently, the input–output transfer function ***A_f_***(s) of the overall feedback structure of Black’s model with the ***γ***-block included would be defined as:(12)$$A_{f}\left(s\right)=\frac{x_{o}\left(s\right)}{x_{s}\left(s\right)}=\frac{A\left(s\right)\gamma \left(s\right)}{1+A\left(s\right)\beta \left(s\right)}$$

The burden of assignment of the sense variable is left to the person undertaking the feedback analysis. This is described more fully in Section 7 and Section 8.

## 4. Feedback Parameter Isolation Method Using the IF Approach

In this section, a method of identifying the feedback parameters **A**(s) and **β**(s) will be described based on the application of a set of transfer functions defined from the input to the intermediate variables of the feedback circuit, such as $x_{Err}$(s), $x_{Fdbk}$(s) and $x_{Sen}$(s). Consequently, these transfer functions will be referred to as the intermediate transfer functions, or IFs for short [12,13]. Through another set of IFs, performance issues, such as component sensitivities and noise gain, can be quantified [13]. This is beyond the scope of this paper so it will not be discussed any further. The reader should simply interpret the IFs as transfer functions from the input to the variable of interest.

According to the block diagram of Figure 7, the **A**(s) and **β**(s) blocks can be defined as the ratio of two signal variables as:(13)$$A\left(s\right)=\frac{x_{Sen}\left(s\right)}{x_{Err}\left(s\right)}$$
and
(14)$$\beta \left(s\right)=\frac{x_{Fdbk}\left(s\right)}{x_{Sen}\left(s\right)}$$

Finally, the ***γ***-block of Figure 7 can be defined as:(15)$$\gamma \left(s\right)=\frac{x_{o}\left(s\right)}{x_{Sen}\left(s\right)}.$$

To understand the physical significance of the ratio of these feedback variables, consider the transfer functions from the input forcing function (V or I) to the intermediate variables associated with the single-loop feedback structure. Rather than tracking all possible variables, we will use *x* with a subscript to represent the circuit variable of interest to stay as general as possible. There are three circuit variables that we mention in the description of the single-loop feedback structure. Specifically, $x_{Err}$(s), $x_{Fdbk}$(s) and $x_{Sen}$(s). However, $x_{Err}$(s) is dependent on the difference between $x_{S}$(s) and $x_{Fdbk}$(s), thus only two transfer functions are necessary to complete this analysis. These are:(16)$$T_{Fdbk}\left(s\right)≜\frac{x_{Fdbk}\left(s\right)}{x_{s}\left(s\right)}$$
and
(17)$$T_{Sen}\left(s\right)≜\frac{x_{Sen}\left(s\right)}{x_{s}\left(s\right)}.$$

The transfer function from the input to the error signal can be defined as:(18)$$T_{Err}\left(s\right)≜\frac{x_{Err}\left(s\right)}{x_{s}\left(s\right)}=1-T_{Fdbk}\left(s\right).$$

Figure 8 depicts the IFs, $T_{Fdbk}$(s), $T_{Err}$(s) and $T_{Sen}$(s), as superimposed on the single-loop feedback structure. Using the above IFs, the feedback components, **A**(s) and **β**(s), can then be expressed as follows:(19)$$A\left(s\right)=\frac{x_{Sen}\left(s\right)}{x_{Err}\left(s\right)}=\frac{T_{Sen}\left(s\right)}{T_{Err}\left(s\right)}$$
and
(20)$$\beta \left(s\right)=\frac{x_{Fdbk}\left(s\right)}{x_{Sen}\left(s\right)}=\frac{T_{Fdbk}\left(s\right)}{T_{Sen}\left(s\right)}.$$

Finally, the ***γ***-block can be identified as:(21)$$\gamma \left(s\right)=\frac{x_{o}\left(s\right)}{x_{Sen}\left(s\right)}=\frac{T_{o}\left(s\right)}{T_{Sen}\left(s\right)}$$
where $T_{o}$(s) is the input–output transfer function being:(22)$$T_{o}\left(s\right)≜\frac{x_{o}\left(s\right)}{x_{s}\left(s\right)}.$$

It is important to point out here that the method of IFs is exact. There are no approximations made to identify **A**(s), **β**(s) or γ(s).

**Example** **1.**
*Consider the unity-gain amplifier configuration involving an op-amp shown in*
Figure 9
*a. The small-signal model of the op-amp is provided in*
Figure 9
*b. Regardless of the complexity of the circuit selected, the IF analysis principles apply to any linear, time-invariant, lumped-element circuit.*


Assuming the sense signal is the output voltage of the amplifier, i.e., $v_{Sen}=v_{o}$, and further, with the feedback signal $v_{Fdbk}$ set equal to the output voltage $v_{o}$, as there is a direct connection between the negative input terminal of the op-amp and its output, the error signal can be easily identified. For our formulation to be compliant with Black’s topology, the error signal $v_{Err}$ must be equal to the difference between the input signal $v_{s}$ and the feedback signal $v_{Fdbk}$, as displayed on the circuit in Figure 9a. Consequently, the feedback parameters **A**(s) and **β**(s) can be computed using the following voltage ratios:(23)$$A\left(s\right)={\left.\frac{T_{Sen}\left(s\right)}{T_{Err}\left(s\right)}\right|}_{\gamma =1}=\frac{T_{o}\left(s\right)}{T_{Err}\left(s\right)}=\frac{T_{o}\left(s\right)}{1-T_{o}\left(s\right)}.$$
and
(24)$$\beta \left(s\right)=\frac{T_{Fdbk}\left(s\right)}{T_{Sen}\left(s\right)}=1$$
as $T_{Fdbk}\left(s\right)=T_{Sen}\left(s\right)=T_{o}\left(s\right)$ and $T_{Err}\left(s\right)=1-T_{Fdbk}\left(s\right)$. Further, $\gamma \left(s\right)=1$, as $v_{Sen}=v_{o}$.

Through a hand analysis, the input–output transfer function was found to be
(25)$$T_{o}\left(s\right)=\frac{C_{in,d}R_{in,d}R_{o}s+R_{in,d}\mu \left(s\right)+R_{o}}{\left(C_{in,d}R_{in,d}R_{s}+C_{in,d}R_{in,d}R_{o}\right)s+R_{in,d}\mu \left(s\right)++R_{s}+R_{o}+R_{in,d}}$$
allowing one to find from Equation (23),
(26)$$A\left(s\right)=\frac{C_{in,d}R_{in,d}R_{o}s+R_{o}+R_{in,d}\mu \left(s\right)}{C_{in,d}R_{in,d}R_{s}s+R_{s}+R_{in,d}}.$$

Now, if we assume a single-pole model for the op-amp, i.e., $\mu \left(s\right)=A_{DC}/\left(1+\frac{s}{\omega_{b}}\right)$ and substituting this into Equation (26), one finds
(27)$$A\left(s\right)=\frac{C_{in,d}R_{in,d}R_{o}s^{2}+\left(C_{in,d}R_{in,d}R_{o}\omega_{b}+R_{o}\right)s+A_{DC}R_{in,d}\omega_{b}+R_{o}\omega_{b}}{\omega_{b}\left(C_{in,d}R_{in,d}R_{s}s+R_{s}+R_{in,d}\right)\left(1+\frac{s}{\omega_{b}}\right)}.$$

Here, it is evident that the A-block has a transfer function that consists of two zeros and two poles. As the zeros were not a part of the op-amp transfer function ***μ***(s), these zeros are caused by the external op-amp impedances. Thus, highlighting the fact that **A**(s) is not necessarily equal to the commonly assumed term of ***μ***(s).

## 5. Noncompliant Single-Loop Negative Feedback Circuits

Circuits that are noncompliant with one of the four single-loop feedback topologies of Figure 2 can still be described with a set of feedback parameters by including a feed-in branch at the front-end with an *α*-block, as shown in Figure 10. Here, a new signal called the reference signal, $x_{Ref}$(s), is included to represent the output signal from the *α*-block. In much the same way as described in Section 4, the IF approach can be used to identify each component of this block diagram [15]. The *α*-block would be identified using the following intermediate transfer function:(28)$$\alpha \left(s\right)=\frac{x_{Ref}\left(s\right)}{x_{S}\left(s\right)}=T_{Ref}\left(s\right)$$
where $T_{Ref}$(s) represents the transfer function from input to the reference signal. The remaining component **A**(s), **β**(s) and ***γ***(s) would be found using the formulas provided in Equations (19)–(21).

The closed-loop input–output signal gain **A*_f_***(s) of the modified single-loop feedback structure expressed in terms of **α**(s), **A**(s), **β**(s) and ***γ***(s) is written as:(29)$$A_{f}\left(s\right)=\frac{x_{o}\left(s\right)}{x_{s}\left(s\right)}=\frac{\alpha \left(s\right)A\left(s\right)\gamma \left(s\right)}{1+A\left(s\right)\beta \left(s\right)}.$$

Now it may appear at first glance as though closed-loop poles would be the roots of the expression $1+A\left(s\right)\beta \left(s\right)=0$. However, the *α*(s) term generally has zeros that cancel with the zeros of the $1+A\left(s\right)\beta \left(s\right)$ term. Thus, the characteristic equation of the modified closed-loop system of Figure 10 must account for this fact, and be written as:(30)$$\frac{1+A\left(s\right)\beta \left(s\right)}{\alpha \left(s\right)\gamma \left(s\right)}=0.$$

An effective loop transmission function can then be defined as:(31)$$A\beta_{eff}\left(s\right)≜\frac{1-\alpha \left(s\right)\gamma \left(s\right)+A\left(s\right)\beta \left(s\right)}{\alpha \left(s\right)\gamma \left(s\right)}$$
where the characteristic equation can be written in the usual form as $1+A\beta_{eff}\left(s\right)=0$. On doing so, the return ratio of this circuit would be equal to $A\beta_{eff}\left(s\right)$ and not the usual quantity **A**(s) × **β**(s). As the dimensions of **A**(s) × **β**(s) are unitless, the units of $A\beta_{eff}\left(s\right)$ take on the dimensions of the inverse of $\alpha \left(s\right)\times \gamma \left(s\right)$.

It is important to note that if ***α***(s) = γ(s) = 1, then the above expressions for the noncompliant feedback structure reduces to the same ones specified for a compliant one.

Example: To illustrate the significance of circuit compliance, consider describing the unity-gain amplifier in Figure 9 with the modified feedback structure in Figure 11. This is done by simply redefining the error signal, as shown in Figure 11, and defining the top terminal of the error signal as the reference signal $v_{Ref}$. Assuming a single-pole model for the op-amp, i.e., $\mu \left(s\right)=A_{DC}/\left(1+\frac{s}{\omega_{b}}\right)$ one can compute the four feedback parameters **α**(s), **A**(s), **β**(s) and ***γ***(s) as follows:(32)$$\alpha \left(s\right)=\frac{\begin{array}{c} C_{in,d}R_{in,d}R_{o}s^{2}+\\ \left(C_{in,d}R_{in,d}R_{o}\omega_{b}+R_{in,d}+R_{o}\right)s\\ +\left(A_{DC}+1\right)R_{in,d}\omega_{b}+R_{o}\omega_{b} \end{array}}{\begin{array}{c} C_{in,d}R_{in,d}\left(R_{o}+R_{s}\right)s^{2}\\ +\left[C_{in,d}R_{in,d}\left(R_{o}+R_{s}\right)\omega_{b}+R_{in,d}+R_{o}+R_{s}\right]s\\ +\left(A_{DC}+1\right)R_{in,d}\omega_{b}+\left(R_{o}+R_{s}\right)\omega_{b} \end{array}}$$
(33)$$A\left(s\right)=\frac{C_{in,d}R_{in,d}R_{o}s^{2}+\left(C_{in,d}R_{in,d}R_{o}\omega_{b}+R_{o}\right)s+A_{DC}R_{in,d}\omega_{b}+R_{o}\omega_{b}}{R_{in,d}s+R_{in,d}\omega_{b}}$$
(34)$$\beta \left(s\right)=1$$
and
(35)$$\gamma \left(s\right)=1$$

Further, the input–output transfer function $\frac{V_{o}}{V_{s}}\left(s\right)$ is found to be:(36)$$\frac{V_{o}}{V_{s}}\left(s\right)=\frac{C_{in,d}R_{in,d}R_{o}s^{2}+\left(C_{in,d}R_{in,d}R_{o}\omega_{b}+R_{o}\right)s+A_{DC}R_{in,d}\omega_{b}+R_{o}\omega_{b}}{\begin{array}{c} C_{in,d}R_{in,d}\left(R_{o}+R_{s}\right)s^{2}+\left[C_{in,d}R_{in,d}\left(R_{o}+R_{s}\right)\omega_{b}+R_{in,d}+R_{o}+R_{s}\right]s\\ +\left(A_{DC}+1\right)R_{in,d}\omega_{b}+\left(R_{o}+R_{s}\right)\omega_{b} \end{array}}$$

It is easy to show that the characteristic equation of $\frac{V_{o}\left(s\right)}{V_{s}\left(s\right)}$ is indeed equal to $\frac{1+A\left(s\right)\beta \left(s\right)}{\alpha \left(s\right)}=0$. The results are straightforward to confirm but lengthy; so these details are not shown here. Another example is provided in Section 6 with many more mathematical details

This section demonstrates the importance of a circuit being topologically compliant, otherwise, the feedback description requires additional terms and a more complicated loop transmission function definition. The simplicity of the feedback theory proposed by Black, combined with the Nyquist stability criterion, would be lost.

## 6. Comparing IF Analysis Isolation Method with the Return Ratio Approach

To demonstrate the interplay between compliant and noncompliant single-loop feedback circuits and the concept of a return ratio, consider the 2nd order lowpass filter circuit of Figure 12a. This circuit uses an internal voltage amplifier with voltage gain *K*. This is equivalent to a voltage-controlled voltage source with gain *K*. To extract the return ratio, the injection method of Middlebrook [2] will be used. Specifically, the dependent voltage source related to the VCVS is replaced by an independent voltage source $v_{t}$, as shown in Figure 12b, and the signal that returns to the input of the VCVS, identified as $v_{r}$, is found, allowing one to write the return ratio as:(37)$$RR\left(s\right)=-\frac{v_{r}}{v_{t}}$$

Using circuit analysis, one can find the return ratio as:(38)$$RR\left(s\right)=-\frac{KC_{1}R_{1}R_{3}s}{\left(C_{1}C_{2}R_{1}R_{2}R_{3}\right)s^{2}+\left(\begin{array}{c} C_{1}R_{1}R_{2}+C_{1}R_{1}R_{3}+\\ C_{2}R_{1}R_{3}+C_{2}R_{2}R_{3} \end{array}\right)s+R_{1}+R_{2}+R_{3}}$$

Assuming the circuit of Figure 12a exhibits a current-mixing/voltage-sampling topology where the feedback current mixes with the input current signal, the input voltage source must first be converted to a current source using the Norton transformation. The resulting circuit is shown in Figure 12c. As this topology is noncompliant with that of Black, the source resistance *R*_1_ is moved to the right-hand side of the current summing node, as shown in Figure 12d, so that the resulting circuit is compliant. Using Equations (13) and (14), the feedback parameters are found to be:(39)$$A\left(s\right)=\frac{v_{Sen}\left(s\right)}{i_{Err}\left(s\right)}=\frac{KR_{1}R_{3}}{\left(C_{2}R_{1}R_{3}+C_{2}R_{2}R_{3}\right)s+R_{1}+R_{2}+R_{3}}$$
and
(40)$$\beta \left(s\right)=\frac{i_{Fdbk}\left(s\right)}{v_{Sen}\left(s\right)}=\frac{\left(C_{1}C_{2}R_{2}R_{3}\right)s^{2}+\left(C_{1}R_{2}+C_{1}R_{3}-KC_{1}R_{3}\right)s}{KR_{3}}$$

Consequently, the product of **A**(s) × **β**(s), can be written as:(41)$$A\left(s\right)\times \beta \left(s\right)=\frac{\left(C_{1}C_{2}R_{1}R_{2}R_{3}\right)s^{2}+\left(C_{1}R_{1}R_{2}+C_{1}R_{1}R_{3}-KC_{1}R_{1}R_{3}\right)s}{\left(C_{2}R_{1}R_{3}+C_{2}R_{2}R_{3}\right)s+R_{1}+R_{2}+R_{3}}$$

On comparing ***RR***(s) with **A**(s) × **β**(s), clearly, they are different. Nonetheless, the sum of their numerator and denominator terms are the same, as is evident in the top two data sets of Table 1. Consequently, the closed-loop poles derived from either ***RR***(s) or **A**(s) × **β**(s) would lead to the same result. 

It is interesting to note that the feedback parameters extracted from the noncompliant circuit of Figure 12c would lead one to a very different set of closed-loop poles. Specifically, parameters **A**(s) and **β**(s) would be found as follows:(42)$$A\left(s\right)=\frac{v_{Sen}\left(s\right)}{i_{Err}\left(s\right)}=\frac{KR_{3}}{C_{2}R_{3}s+1}$$
and
(43)$$\beta \left(s\right)=\frac{i_{Fdbk}\left(s\right)}{v_{Sen}\left(s\right)}=\frac{\left(C_{1}C_{2}R_{2}R_{3}\right)s^{2}+\left(C_{1}R_{2}+C_{1}R_{3}-KC_{1}R_{3}\right)s}{KR_{3}}$$

Further, the product of **A**(s) × **β**(s) would then be:(44)$$A\left(s\right)\times \beta \left(s\right)=\frac{\left(C_{1}C_{2}R_{2}R_{3}\right)s^{2}+\left(C_{1}R_{2}+C_{1}R_{3}-KC_{1}R_{3}\right)s}{C_{2}R_{3}s+1}$$

As this product is quite different than the ones found from the compliant circuit, it is not surprising that the sum of the numerator and denominator terms are also quite different. These are shown in the third data set of Table 1. However, if the effective loop transmission function $A\beta_{eff}\left(s\right)$ is used instead for the noncompliant circuit, then one would find the sum of the numerator and denominator terms the same.

To see this, **α**(s) and ***γ***(s) were found from the noncompliant circuit of Figure 12c according to Equations (21) and (28) as follows:(45)$$\alpha \left(s\right)=\frac{\left(C_{1}C_{2}R_{1}R_{2}R_{3}\right)s^{2}+\left(C_{1}R_{1}R_{2}+C_{1}R_{1}R_{3}-KC_{1}R_{1}R_{3}\right)+R_{1}}{\left(C_{1}C_{2}R_{1}R_{2}R_{3}\right)s^{2}+\left(\begin{array}{c} -KC_{1}R_{1}R_{3}s+C_{1}R_{1}R_{2}+C_{1}R_{1}R_{3}\\ +C_{2}R_{1}R_{3}+C_{2}R_{2}R_{3} \end{array}\right)s+R_{1}+R_{2}+R_{3}}$$
and
(46)$$\gamma \left(s\right)=1$$

Evaluating $A\beta_{eff}\left(s\right)$ according to Equation (32), one gets
(47)$$\begin{array}{c} A\beta_{eff}\left(s\right)\\ =\frac{\left(C_{1}C_{2}R_{1}R_{2}R_{3}\right)s^{2}+\left(C_{1}R_{1}R_{2}+C_{1}R_{1}R_{3}-KC_{1}R_{1}R_{3}+C_{2}R_{2}R_{3}\right)s+R_{2}+R_{3}}{C_{2}R_{1}R_{3}s+R_{1}} \end{array}$$

Here, the sum of its numerator and denominator terms are indeed the same as those given by the return ratio or the loop transmission function derived from its compliant circuit configuration.

## 7. Selecting Mixing and Sensing Signals of Complex Single-Loop Circuits

To illustrate the process of selecting the mixing and sensing variables of a complex circuit involving a single-loop feedback arrangement, consider the two-stage BJT amplifier circuit shown in Figure 13a. Here, the amplifier is driven with a voltage source $v_{s}$, and the output of the amplifier is a node voltage designated as $v_{o}$. A feedback path from the emitter of Q_2_ to the base of Q_1_ involving *R_f_* can easily be identified. In general, single-loop feedback circuits have only a single feedback loop, so identifying it is rather self-evident. The rest of the loop is made up of a cascade of two single-stage amplifiers involving Q_1_ and Q_2_. The entire loop is highlighted with a red ellipse superimposed on the schematic shown in Figure 13a. As the output signal $v_{o}$ does not lie on the circuit path involving the feedback loop, this signal cannot be sensed by the feedback loop. Consequently, another signal must be chosen as the sense signal, and it must lie on the circuit path of the feedback loop. It can be another node voltage, say, for instance, the emitter voltage of Q_2_, the collector of Q_1_, etc., or as a branch current, such as the emitter or base current of Q_2_. For the sake of our discussion, the emitter current of Q_2_ is identified as the sense signal as shown in Figure 13b.

Our next step is to identify the front-end mixing signals. There are always two options to consider, voltage or current mixing. If we assume voltage mixing, then the node that terminates the feedback connection on the input side of the amplifier could be designated as the feedback voltage signal $v_{Fdbk}$. Corresponding, to be compliant with Black’s mixing arrangement, then the difference between the input voltage signal $v_{s}$ and the feedback signal $v_{Fdbk}$ must be designated as the error voltage signal, $v_{Err}$, as shown in Figure 13c.

Conversely, one can assume that the node terminating the feedback on the input side is the current mixing (or summing) node. As such, the branch current in the feedback connection can be identified as the feedback current $i_{Fdbk}$. To be compliant with Black’s current-mixing arrangement, the input voltage source must be converted to a current source through a Norton transformation. The source resistance *R_S_* must also be moved to the right side of the current summing node, as demonstrated in the schematic of Figure 13d. To avoid disturbing the DC bias levels of the transistors, the DC blocking capacitor *C_C1_* should be placed in series with R_S_.

Both circuits of Figure 13c,d are compliant with Black’s topology. An IF analysis can be performed and the feedback parameters **A**(s) and **β**(s) can be identified. In addition, as the output signal is not the same as the sense variable, the gamma term ***γ***(s) would also be required to fully described the input–output operation of these two circuits.

## 8. Alternative Forms of Black’s Feedback Representations

The feedback parameters $A\left(s\right)$ and $\beta \left(s\right)$ derived from a complaint single-loop feedback circuit are not unique. The expressions of $A\left(s\right)$ and $\beta \left(s\right)$ will depend on several factors:(1)The way the feedback signal is mixed with the input signal, i.e., voltage or current mixing;(2)The designation of which signal is the feedback signal, and which is the error signal, and;(3)The signal that is being sensed by the feedback loop, i.e., voltage at a node or a current through some branch in the feedback loop.

This suggests that there are numerous possible combinations of the feedback variables. As all formulations are derived from the same closed-loop circuit, they will all have the same poles. Therefore, as noted earlier, this suggests that the sum of the numerator and denominator polynomials of the product of **A**(s) × **β**(s) will be the same for each formulation, i.e.,
(48)$$num\left\{{\left.A\left(s\right)\times \beta \left(s\right)\right|}_{1}\right\}+den\left\{{\left.A\left(s\right)\times \beta \left(s\right)\right|}_{1}\right\}=\;num\left\{{\left.A\left(s\right)\times \beta \left(s\right)\right|}_{2}\right\}+den\left\{{\left.A\left(s\right)\times \beta \left(s\right)\right|}_{2}\right\}=\;\vdots \;num\left\{{\left.A\left(s\right)\times \beta \left(s\right)\right|}_{N}\right\}+den\left\{{\left.A\left(s\right)\times \beta \left(s\right)\right|}_{N}\right\}$$
assuming there are *N* different combinations of the feedback variables chosen from the same compliant single-loop feedback circuit.

To demonstrate this invariance, together with the fact that the feedback parameters **A**(s) and **β**(s) will vary with each feedback arrangement, consider the single-stage common-emitter (CE) BJT amplifier with resistive feedback shown in Figure 14. Here, there are four different assignments of the feedback variables for the same circuit. In part (a), the circuit variables are assigned based on being compliant with a voltage-mixing/voltage-sensing topology. The sense variable is assigned as the collector voltage of Q_1_. In part (b), a Norton transformation is performed on the input voltage source to provide an input current excitation. The source resistance is moved towards to the base of Q_1_ so that the feedback variable assignment would be compliant with a current-mixing/voltage-sensing topology. In part (c), a voltage-mixing/voltage-sensing topology is used; however, in this case, the roles of the feedback and error voltages are reversed from the situation depicted in part (a). This example is to highlight the duality of the two variables in a single-loop feedback circuit. In part (d), a voltage-mixing arrangement identical to part (a) is used, however, the collector current is to act as the sense variable rather than the collector voltage. Thus, a voltage-mixing/current-sensing topology has been selected.

Assuming a hybrid-pi model for the BJT transistor with only C*_μ_* included, the feedback parameters for all four circuit arrangements are computed using the Maple symbolic analysis tool. Only the C*_μ_* capacitance of the hybrid-pi model is used here to keep the length of the polynomials associated with the feedback parameters manageable. Including C*_μ_* in this analysis will not alter our observation, as this was verified independently. The results of this analysis are summarized in Table 2 for all four topologies for the amplifier circuit of Figure 14. While the feedback parameters are all different, as well as their units, what is important to recognize here is that the sum of the numerator and denominator polynomials for each formulation are all the same. Thus, confirming invariant properties of the many possible formulations of the same single-loop feedback circuit.

## 9. Conclusions

A method to identify the feedback parameters **A**(s) and **β**(s) of a single-loop feedback topology as defined by Black has been proposed. The method makes use of the concept of intermediate transfer functions (IFs) and provides the circuit conditions on which the method applies. No special algorithms are required, only access to a computer-based circuit analysis tool, such as Spice, or if simple enough, hand analysis. If certain circuit conditions do not hold, an alternative feedback formulation was proposed that requires a different loop transmission function than that proposed by Black. This is the first time a method has been proposed that can identify the feedback parameters of a single-loop feedback circuit in terms of its circuit components. The proposed approach does not involve the concept of a return ratio to do so. While the examples used in this paper were relatively simple, as the intent was to easily track the various IF transfer functions, the proposed method is general and applicable to any single-loop feedback circuit.

## Figures and Tables

**Figure 1 sensors-22-04303-f001:**
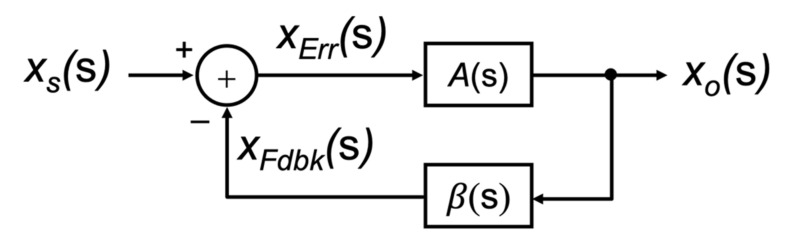
The general form of a negative feedback structure, as first proposed by H. Black.

**Figure 2 sensors-22-04303-f002:**
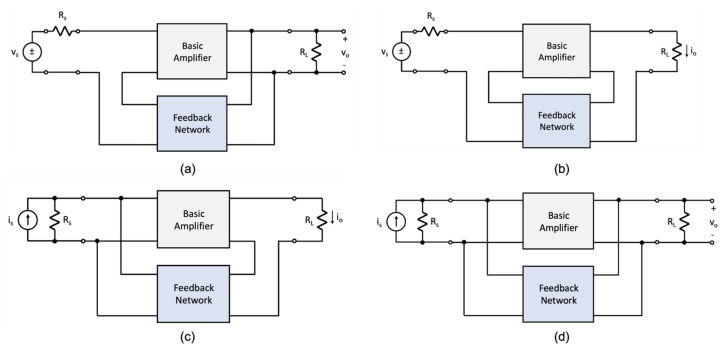
The four noncompliant single-loop feedback topologies incorporated with circuits: (**a**) voltage-mixing/voltage-sensing, (**b**) voltage-mixing/current-sensing, (**c**) current-mixing/current-sensing, and (**d**) current-mixing/voltage-sensing.

**Figure 3 sensors-22-04303-f003:**
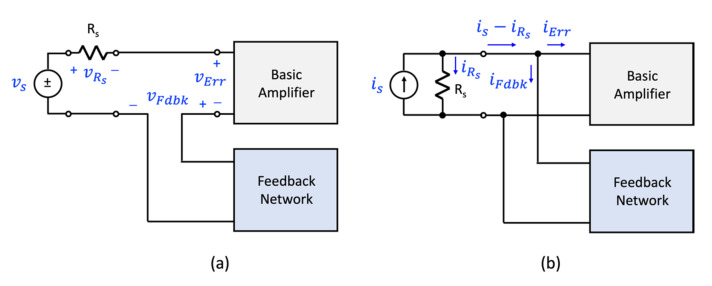
Highlighting problem with voltage and current mixing when a source resistance is present. (**a**) voltage mixing, and (**b**) current mixing.

**Figure 4 sensors-22-04303-f004:**
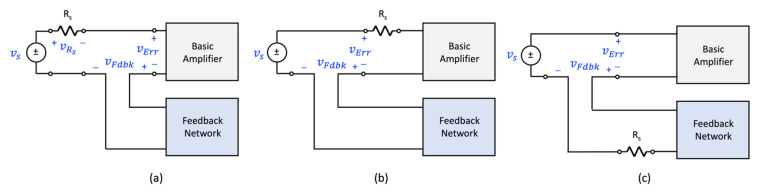
Three equivalent voltage-mixing arrangements; two are compliant with Black’s single-loop feedback structure: (**a**) noncompliant, (**b**) compliant, and (**c**) compliant.

**Figure 5 sensors-22-04303-f005:**
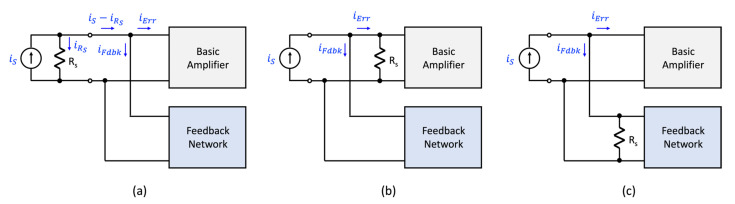
Three equivalent current-mixing arrangements; two are compliant with Black’s single-loop feedback structure: (**a**) noncompliant, (**b**) compliant option 1, and (**c**) compliant option 2.

**Figure 6 sensors-22-04303-f006:**
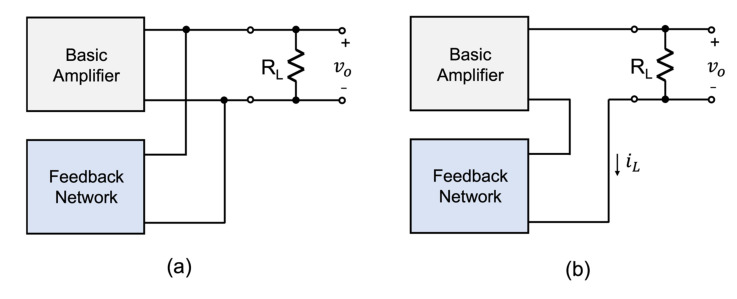
Highlighting the physical difference between an output signal from a circuit with a feedback loop and the signal being sensed by the feedback network. (**a**) The output voltage is the same signal that is being sensed by the feedback network, and (**b**) the output voltage is different from the current signal that is being sensed by the feedback network.

**Figure 7 sensors-22-04303-f007:**
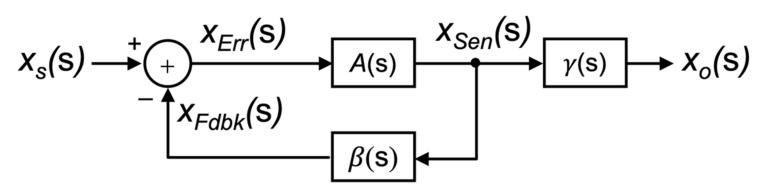
Including a ***γ***(s)-block that relates the sense variable ***x_Sen_*** to the designated output signal ***x*_o_**.

**Figure 8 sensors-22-04303-f008:**
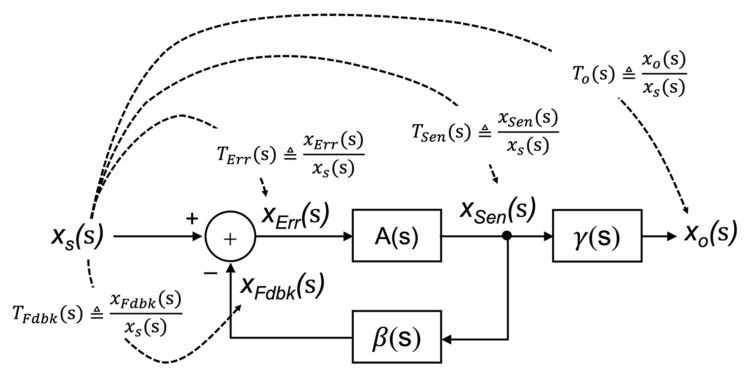
Illustrating the intermediate transfer functions associated with a single-loop feedback circuit.

**Figure 9 sensors-22-04303-f009:**
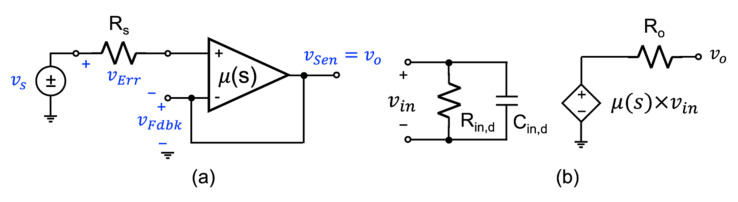
Voltage-mixing/voltage-sensing topology: (**a**) Unity-gain amplifier using an op-amp, and (**b**) op-amp circuit model with general gain function *μ*(s). The voltage-mixing signal are highlighted in blue.

**Figure 10 sensors-22-04303-f010:**
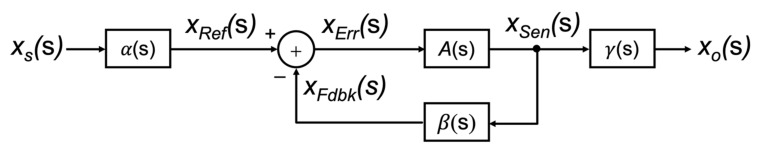
Inclusion of a feed-in branch **α**(s) to expand the circuit range of applicability of a single-loop negative feedback system description.

**Figure 11 sensors-22-04303-f011:**
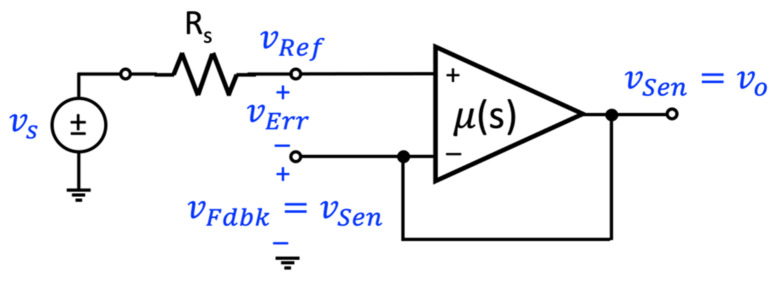
A unity-gain amplifier circuit that is to be mapped to the modified single-loop feedback structure of Figure 10. The circuit is the same, but the voltage-mixing variables have been changed.

**Figure 12 sensors-22-04303-f012:**
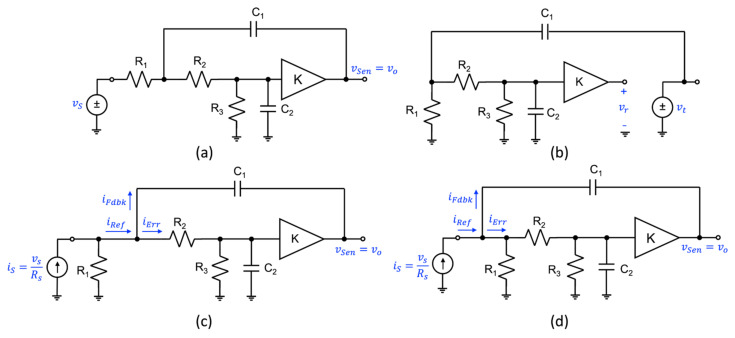
Comparing noncompliant, compliant feedback topologies with the Middlebrook loop injection method for extracting its return ratio: (**a**) voltage-input voltage-output active filter circuit, (**b**) evaluating the return ratio of the active filter circuit using Middlebrook’s injection method by replacing the dependent voltage source related to the VCVS with an independent voltage source, (**c**) Norton equivalent circuit representation that is noncompliant with Black’s feedback topology, and (**d**) Norton equivalent circuit representation that is compliant.

**Figure 13 sensors-22-04303-f013:**
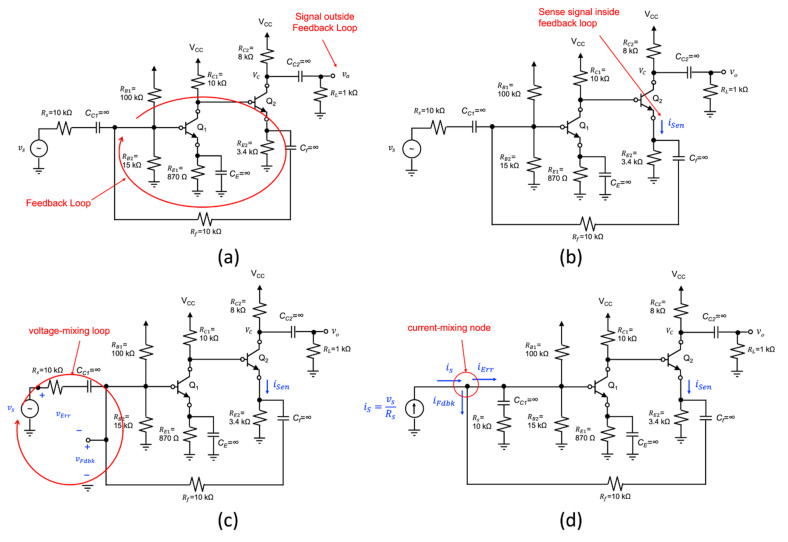
Preparing a current-mixing/current-sensing feedback circuit for feedback parameters isolation: (**a**) Identifying the feedback loop. (**b**) As the output voltage is outside the feedback loop of the amplifier, a sensing current has been identified that is inside the loop. (**c**) Identifying the system variables that form a voltage-mixing loop that is compliant with Black’s topology. (**d**) A fully compliant circuit arrangement that meets the definition of a current-mixing/current-sensing topology.

**Figure 14 sensors-22-04303-f014:**
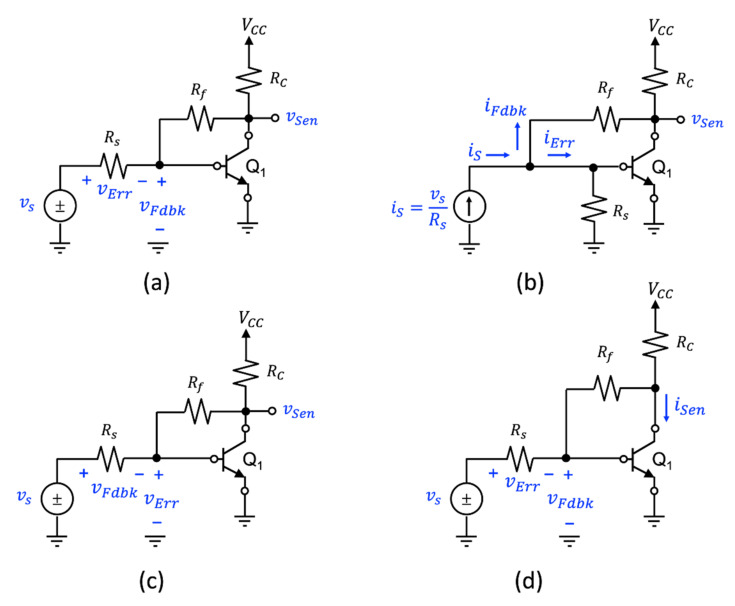
Different feedback formulation for a CE BJT amplifier with resistive feedback: (**a**) voltage-mixing/voltage-sensing compliant arrangement with feedback variable highlighted in blue, (**b**) through a Norton transformation, the CE amplifier is rearranged into an equivalent current-mixing/voltage-sensing compliant topology, (**c**) interchanging the feedback and error signal designations, and (**d**) selecting the sense signal as the collector current rather than the collector voltage.

**Table 1 sensors-22-04303-t001:** Numerator and Denominator Terms and their Combined Sums for the Circuits of Figure 12. (Green shaded row indicates agreement; red shaded row indicates disagreement).

$RR\left(s\right)$ For Ciruit of Figure 12b	
$num\left\{RR\left(s\right)\right\}$	$-KC_{1}R_{1}R_{3}s$
$den\left\{RR\left(s\right)\right\}$	$\left(C_{1}C_{2}R_{1}R_{2}R_{3}\right)s^{2}+\left(C_{1}R_{1}R_{2}+C_{1}R_{1}R_{3}+C_{2}R_{1}R_{3}+C_{2}R_{2}R_{3}\right)s+R_{1}+R_{2}+R_{3}$
$num\left\{RR\left(s\right)\right\}+den\left\{RR\left(s\right)\right\}$	$\left(C_{1}C_{2}R_{1}R_{2}R_{3}\right)s^{2}+\left(C_{1}R_{1}R_{2}+C_{1}R_{1}R_{3}+C_{2}R_{1}R_{3}+C_{2}R_{2}R_{3}-KC_{1}R_{1}R_{3}\right)s+R_{1}+R_{2}+R_{3}$
$A\left(s\right)\times \beta \left(s\right)\;$ **For Compliant Circuit of Figure 12d**	
$num\left\{A\left(s\right)\times \beta \left(s\right)\right\}$	$\left(C_{1}C_{2}R_{1}R_{2}R_{3}\right)s^{2}+\left(C_{1}R_{1}R_{2}+C_{1}R_{1}R_{3}-KC_{1}R_{1}R_{3}\right)$s
$den\left\{A\left(s\right)\times \beta \left(s\right)\right\}$	$\left(C_{2}R_{1}R_{3}+C_{2}R_{2}R_{3}\right)s+R_{1}+R_{2}+R_{3}$
$num\left\{A\left(s\right)\times \beta \left(s\right)\right\}+den\left\{A\left(s\right)\times \beta \left(s\right)\right\}$	$\left(C_{1}C_{2}R_{1}R_{2}R_{3}\right)s^{2}+\left(C_{1}R_{1}R_{2}+C_{1}R_{1}R_{3}+C_{2}R_{1}R_{3}+C_{2}R_{2}R_{3}-KC_{1}R_{1}R_{3}\right)s+R_{1}+R_{2}+R_{3}$
$A\left(s\right)\times \beta \left(s\right)\;$ **For Noncompliant Circuit of Figure 12c**	
$num\left\{A\left(s\right)\times \beta \left(s\right)\right\}$	$\left(C_{1}C_{2}R_{2}R_{3}\right)s^{2}+\left(C_{1}R_{2}+C_{1}R_{3}-KC_{1}R_{3}\right)s$
$den\left\{A\left(s\right)\times \beta \left(s\right)\right\}$	$C_{2}R_{3}s+1$
$num\left\{A\left(s\right)\times \beta \left(s\right)\right\}+den\left\{A\left(s\right)\times \beta \left(s\right)\right\}$	$\left(C_{1}C_{2}R_{2}R_{3}\right)s^{2}+\left(C_{1}R_{2}+C_{1}R_{3}+C_{2}R_{3}s-KC_{1}R_{3}\right)s+1$
$A\beta_{eff}\left(s\right)$ **For Noncompliant Circuit of Figure 12c**	
$num\left\{A\beta_{eff}\left(s\right)\right\}$	$\left(C_{1}C_{2}R_{1}R_{2}R_{3}\right)s^{2}+\left(C_{1}R_{1}R_{2}+C_{1}R_{1}R_{3}+C_{2}R_{2}R_{3}-KC_{1}R_{1}R_{3}\right)+R_{2}+R_{3}$
$den\left\{A\beta_{eff}\left(s\right)\right\}$	$C_{2}R_{1}R_{3}s+R_{1}$
$num\left\{A\beta_{eff}\left(s\right)\right\}+den\left\{A\beta_{eff}\left(s\right)\right\}$	$\left(C_{1}C_{2}R_{1}R_{2}R_{3}\right)s^{2}+\left(C_{1}R_{1}R_{2}+C_{1}R_{1}R_{3}+C_{2}R_{1}R_{3}+C_{2}R_{2}R_{3}-KC_{1}R_{1}R_{3}\right)s+R_{1}+R_{2}+R_{3}$

**Table 2 sensors-22-04303-t002:** Feedback parameters **A**(s) and **β**(s), and their combined sums for the circuits of Figure 14 (Green shaded row indicates agreement).

$A\left(s\right)\;\mathrm{and}\;\beta \left(s\right)$ For Circuit of Figure 14a	
$A\left(s\right)\left[\frac{V}{V}\right]$	$\frac{r_{\pi }\left(C_{\mu }R_{C}R_{f}r_{o}s-g_{m}r_{o}R_{C}R_{f}+R_{C}r_{o}\right)}{\begin{array}{c} C_{\mu }\left(g_{m}r_{o}R_{C}R_{s}R_{f}r_{\pi }+r_{o}R_{C}R_{s}R_{f}+r_{\pi }R_{C}R_{s}R_{f}+R_{f}R_{s}r_{o}r_{\pi }\right)s\\ +g_{m}r_{o}R_{C}R_{s}r_{\pi }+R_{C}R_{f}R_{s}+R_{C}R_{s}r_{o}+R_{C}R_{s}r_{\pi }+R_{f}R_{s}r_{o}+R_{s}r_{o}r_{\pi } \end{array}}$
$\beta \left(s\right)\;\left[\frac{V}{V}\right]$	$\frac{C_{\mu }R_{C}R_{f}r_{o}s+R_{C}R_{f}+R_{C}r_{o}+R_{f}r_{o}}{C_{\mu }R_{C}R_{f}r_{o}s-g_{m}r_{o}R_{C}R_{f}+R_{C}r_{o}}$
$num\left\{A\left(s\right)\times \beta \left(s\right)\right\}+$ $den\left\{A\left(s\right)\times \beta \left(s\right)\right\}$	$C_{\mu }\left(g_{m}r_{o}R_{C}R_{s}R_{f}r_{\pi }+r_{o}R_{C}R_{s}R_{f}+r_{\pi }R_{C}R_{s}R_{f}+R_{C}R_{f}r_{o}r_{\pi }+R_{f}R_{s}r_{o}r_{\pi }\right)s+$ $g_{m}r_{o}R_{C}R_{s}r_{\pi }+R_{C}R_{f}R_{s}+R_{C}R_{f}r_{\pi }+R_{C}R_{s}r_{o}+R_{C}R_{s}r_{\pi }+R_{C}r_{o}r_{\pi }+R_{f}R_{s}r_{o}+R_{f}r_{o}r_{\pi }+R_{s}r_{o}r_{\pi }$
$A\left(s\right)\;and\;\beta \left(s\right)\;$ **For Circuit of Figure 14b**	
$A\left(s\right)\;\left[\frac{V}{A\;}\right]$	$\frac{R_{s}r_{\pi }\left(C_{\mu }R_{C}R_{f}r_{o}s-g_{m}r_{o}R_{C}R_{f}+R_{C}r_{o}\right)}{\begin{array}{c} C_{\mu }\left(g_{m}r_{o}R_{C}R_{s}R_{f}r_{\pi }+r_{o}R_{C}R_{s}R_{f}+r_{\pi }R_{C}R_{s}R_{f}+r_{o}R_{C}R_{f}r_{\pi }+R_{f}R_{s}r_{o}r_{\pi }\right)s\\ +R_{C}R_{f}R_{s}+R_{C}R_{f}r_{\pi }+R_{C}R_{s}r_{o}+R_{C}r_{o}r_{\pi }+R_{f}R_{s}r_{o}+R_{f}r_{o}r_{\pi } \end{array}}$
$\beta \left(s\right)\;\left[\frac{A}{V}\right]$	$\frac{R_{C}g_{m}r_{o}+R_{C}+r_{o}}{C_{\mu }R_{C}R_{f}r_{o}s-g_{m}r_{o}R_{C}R_{f}+R_{C}r_{o}}$
$num\left\{A\left(s\right)\times \beta \left(s\right)\right\}+$ $den\left\{A\left(s\right)\times \beta \left(s\right)\right\}$	$C_{\mu }\left(g_{m}r_{o}R_{C}R_{s}R_{f}r_{\pi }+r_{o}R_{C}R_{s}R_{f}+r_{\pi }R_{C}R_{s}R_{f}+R_{C}R_{f}r_{o}r_{\pi }+R_{f}R_{s}r_{o}r_{\pi }\right)s+$ $g_{m}r_{o}R_{C}R_{s}r_{\pi }+R_{C}R_{f}R_{s}+R_{C}R_{f}r_{\pi }+R_{C}R_{s}r_{o}+R_{C}R_{s}r_{\pi }+R_{C}r_{o}r_{\pi }+R_{f}R_{s}r_{o}+R_{f}r_{o}r_{\pi }+R_{s}r_{o}r_{\pi }$
$A\left(s\right)\;and\;\beta \left(s\right)\;$ **For Circuit of Figure 14c**	
$A\left(s\right)\;\left[\frac{V}{V}\right]$	$\frac{R_{C}\left(C_{\mu }R_{f}r_{o}s-g_{m}r_{o}R_{f}+r_{o}\right)}{C_{\mu }R_{C}R_{f}r_{o}s+R_{C}R_{f}+R_{C}r_{o}+R_{f}r_{o}}$
$\beta \left(s\right)\;\left[\frac{V}{V}\right]$	$\frac{\begin{array}{c} C_{\mu }\left(g_{m}r_{o}R_{C}R_{s}R_{f}r_{\pi }+r_{o}R_{C}R_{s}R_{f}+r_{\pi }R_{C}R_{s}R_{f}+R_{f}R_{s}r_{o}r_{\pi }\right)s\\ +g_{m}r_{o}R_{C}R_{s}r_{\pi }+R_{C}R_{f}R_{s}+R_{C}R_{s}r_{o}+R_{C}R_{s}r_{\pi }+R_{f}R_{s}r_{o}+R_{s}r_{o}r_{\pi } \end{array}}{r_{\pi }\left(C_{\mu }R_{C}R_{f}r_{o}s-g_{m}r_{o}R_{C}R_{f}+R_{C}r_{o}\right)}$
$num\left\{A\left(s\right)\times \beta \left(s\right)\right\}+$ $den\left\{A\left(s\right)\times \beta \left(s\right)\right\}$	$C_{\mu }\left(g_{m}r_{o}R_{C}R_{s}R_{f}r_{\pi }+r_{o}R_{C}R_{s}R_{f}+r_{\pi }R_{C}R_{s}R_{f}+R_{C}R_{f}r_{o}r_{\pi }+R_{f}R_{s}r_{o}r_{\pi }\right)s+$ $g_{m}r_{o}R_{C}R_{s}r_{\pi }+R_{C}R_{f}R_{s}+R_{C}R_{f}r_{\pi }+R_{C}R_{s}r_{o}+R_{C}R_{s}r_{\pi }+R_{C}r_{o}r_{\pi }+R_{f}R_{s}r_{o}+R_{f}r_{o}r_{\pi }+R_{s}r_{o}r_{\pi }$
$A\left(s\right)\;and\;\beta \left(s\right)\;$ **For Circuit of Figure 14d**	
$A\left(s\right)\;\left[\frac{A}{V\;}\right]$	$\frac{r_{\pi }\left(-C_{\mu }R_{f}r_{o}s+g_{m}r_{o}R_{C}+g_{m}r_{o}R_{f}+R_{C}\right)}{\begin{array}{c} C_{\mu }\left(g_{m}r_{o}R_{C}R_{s}R_{f}r_{\pi }+r_{o}R_{C}R_{s}R_{f}+r_{\pi }R_{C}R_{s}R_{f}+R_{f}R_{s}r_{o}r_{\pi }\right)s\\ +g_{m}r_{o}R_{C}R_{s}r_{\pi }+R_{C}R_{f}R_{s}+R_{C}R_{s}r_{o}+R_{C}R_{s}r_{\pi }+R_{f}R_{s}r_{o}+R_{s}r_{o}r_{\pi } \end{array}}$
$\beta \left(s\right)\;\left[\frac{V}{A}\right]$	$\frac{C_{\mu }R_{C}R_{f}r_{o}s+R_{C}R_{f}+R_{C}r_{o}+R_{f}r_{o}}{-C_{\mu }R_{f}r_{o}s+g_{m}r_{o}R_{C}+g_{m}r_{o}R_{f}+R_{C}}$
$num\left\{A\left(s\right)\times \beta \left(s\right)\right\}+$ $den\left\{A\left(s\right)\times \beta \left(s\right)\right\}$	$C_{\mu }\left(g_{m}r_{o}R_{C}R_{s}R_{f}r_{\pi }+r_{o}R_{C}R_{s}R_{f}+r_{\pi }R_{C}R_{s}R_{f}+R_{C}R_{f}r_{o}r_{\pi }+R_{f}R_{s}r_{o}r_{\pi }\right)s+$ $g_{m}r_{o}R_{C}R_{s}r_{\pi }+R_{C}R_{f}R_{s}+R_{C}R_{f}r_{\pi }+R_{C}R_{s}r_{o}+R_{C}R_{s}r_{\pi }+R_{C}r_{o}r_{\pi }+R_{f}R_{s}r_{o}+R_{f}r_{o}r_{\pi }+R_{s}r_{o}r_{\pi }$

## Data Availability

Not applicable.

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
