# Peer review of "Identifying A(s) and β(s) in Single-Loop Feedback Circuits Using the Intermediate Transfer Function Approach"

_sensors, 2022, doi:10.3390/s22114303_

Round 1

Reviewer 1 Report

I enjoyed reading this paper very much. The proposed method for identifying feedback parameters A and Beta is a useful alternative to existing approaches. In addition, the importance of defining circuit variables to achieve "compliance" is explained clearly. Nevertheless, I have several comments and questions below which the author should address before publication.

1. Page 2: The explanation of the need for Eqn. (2) is not clear. How does Eqn. (1) "suggest" Eqn. (2)?

2. Page 6: I find the following statement on the output sensing compliance ambiguous: "...it is paramount that the sense variable be located directly on the circuit path of the feedback loop". How does one ensure that the sense variable chosen conforms to this? While a current variable defined along the feedback path is obvious, it is less clear if defining a node voltage with respect to ground is considered a variable on the feedback path.

3. Page 7: Do intermediate transfer functions (IFs) have special properties? The fact that you call your technique the "IF approach" and have referenced a few papers on IFs seems to suggest this.

4. Page 9, Fig. 8: T_o(s) should have x_s(s) in the denominator, not x_sen(s).

5. Page 10, Fig. 9: The figure caption is incorrect.

6. Page 11, Fig. 10: The wrong figure is shown here (i.e., there's no diagram showing alpha).

7. Page 12: In the example using Fig. 12, it should be defined that scalar K < 0 such that closed-loop stability is achieved when A*Beta > 0.

8. Page 14, Eqn. (42): A(s) should be v_o/i_err, not the ratio of two voltages.

9. Page 14, Eqn. (43): Beta should be in the form of i/v, not v/v.

10. Page 11, Eqn. (31): For certain feedback configurations, it is possible that the units of alpha*gamma are different from the units of A*Beta in the numerator. How do you handle this case?

11. Two-port network and return-ratio analyses both include systematic ways to calculate the effect of the loop gain on the input/output impedance of the closed-loop circuit. Briefly explain how your method could be used to do this.

Reviewer 2 Report

In this paper, the authors present an exact method to uniquely identify each feedback parameter A or ? in terms of the circuit components. Also, identify the circuit conditions for which the product of A × ? leads to the correct closed-loop poles. The proposed method can be performed using any Spice-like program, as no additional tools are required, or simply worked by hand or on a computer using traditional circuit analysis techniques. A distinction is made between single-loop feedback circuits that conform to the structure proposed by Black and those that do not. This article is clear, concise, and suitable for the scope of the journal. Only one small suggestion is supplied:
Suggest the authors supply more detail about the intermediate transfer functions associated with a single-loop feedback circuit.
